# The Spanish Version of the State Self-Compassion Scale–Long Form (SSCS–L): A Study of Its Validity and Reliability in a Sample of Nursing Students

**DOI:** 10.3390/ijerph191610174

**Published:** 2022-08-17

**Authors:** Laura Galiana, Mireia Guillén, Antonia Pades, Sarah L. Flowers, Gabriel Vidal-Blanco, Noemí Sansó

**Affiliations:** 1Department of Methodology for the Behavioral Sciences, University of Valencia, 46010 Valencia, Spain; 2Department of Nursing and Physiotherapy, University of the Balearic Islands, 07122 Palma, Spain; 3Department of Nursing, University of Valencia, 46010 Valencia, Spain; 4Balearic Islands Health Research Institute (IDISBA), 07004 Palma, Spain

**Keywords:** self-compassion, nursing, validation, burnout, well-being

## Abstract

Background: In nurses, self-compassion mitigates the effects of stress, burnout and compassion fatigue, and enhances empathy, compassion and well-being and quality of life. The Self-Compassion Scale is the most-used instrument. The aim of this study is to validate the Spanish version of the new developed State Self-Compassion Scale–Long (SSCS–L). Methods: Students of the first year of the Nursing Degree were surveyed online. Together with the SSCS–L, their levels of positive and negative affect was reported. Analyses included descriptive statistics, competitive confirmatory factor analysis, evidence on criterion-related validity and estimates of reliability. Results: The best fitting model for the SSCS–L was the one hypothesizing six-correlated factors of self-compassion: self-kindness, common humanity, mindfulness, self-judgement, isolation, and over-identification. Positive relations between the positive dimensions of self-compassion and positive affect were found, whereas there were negative relations between the positive poles of self-compassion and negative affect. Estimates of reliability were adequate, except for the dimension of over-identification. Conclusions: Self-compassion has become a key competency for nurses. The SSCS–L is an appropriate tool to allow an adequate assessment of self-compassion in experimental contexts.

## 1. Introduction

Compassion is a multi-faceted construct that can be used to define behavior towards oneself or others. Self-compassion is when compassion is turned inward, approaching one’s inadequacies and failures with kindness [1,2,3], instead of with criticism or self-judgment. It has been defined as an ability to hold one’s feelings with warmth [1,2]. Neff [1,2] further expands on this concept identifying both positive and negative aspects of self-compassion, with self-kindness, common humanity and mindfulness as examples of compassionate behavior, in contrast to self-judgement, isolation and over-identification [4]. It is therefore “a healthy way of relating to oneself in times of suffering, and applies to situations of failure, perceived inadequacy, or general life difficulties” [5] (p.121).

Defined as such, self-compassion has been consistently related to several positive psychological outcomes. For example, literature points its potential to increase life satisfaction [6,7], harmony [6], and subjective well-being [8,9,10]. Indeed, interventions that focus on cultivating self-compassion and its components have been found to reduce shame and depression [11,12,13,14,15]. In line with this research, self-compassion has been also considered a protector against of suicidal thoughts and self-harm [16]. It also benefits interpersonal relationships [17], by improving mindful parenting [18,19,20], decreasing attachment anxiety and avoidance [21,22,23], and improving romantic relationship satisfaction and dyadic adjustment, as well as being better accepting of both our own and our romantic partner’s personal flaws [24,25]. It has also been linked to several improvements in the working environment, increasing levels of job satisfaction and job performance [26,27,28] and improving professional quality of life [29].

This is especially true for nursing professionals and students where self-compassion can be seen as a useful tool to face excessive workloads. Both in qualitative and quantitative terms, nurses are exposed to very high levels of stress caused by the work environment [30,31]. Considering this, self-compassion could be useful in order to face such stress, as it seems to act as a buffer against the effects of stress [32,33,34,35], burnout [32,33,35,36,37,38,39,40,41,42], and compassion fatigue [35,40,41,42]. It has also been linked to higher levels of empathy [35,43], compassion for others [44], compassion satisfaction [35,37,38,41,42,45], increased compassionate care [36,46], and higher caring efficacy [47]. Additionally, adequate levels of self-compassion have been related to well-being [36,40,46] and quality of life [32,40].

The most common tool to measure self-compassionate behaviors is the Self-Compassionate Scale (SCS) [29]. This consists of a 26-part self-reporting questionnaire that measures individual differences with regard to trait and self-compassion. A shortened version of this form has also been developed: the Self-Compassion Scale–Short Form (SCS–SF) [48]. This version consists of only 12 items instead of the original 26. Both the SCS and the SCS–SF have been adapted and translated to several languages, including Brazilian Portuguese [49], Chinese [50], French [51], German [52], Greek [53], Italian [54], Japanese [55], Korean [56], Norwegian [57], Persian [58], Portuguese [59], or Spanish [60]. A recent review shows that the SCS is by far the most used instrument with regards to the study of self-compassion for nurses and midwifes [61], with the Spanish version of the SCS–SF being recently validated [42].

Based on this conceptualization and measurement of self-compassion, Neff et al. [62] have recently developed the State Self-Compassion State (SSCS). As stated by Neff et al. [62], there is a growing tendency to study self-compassion in experimental contexts. Literature on self-compassion shows that it is common to induce a self-compassionate mindset, either with writing exercises or guided meditations [63,64,65,66,67]. In such studies, the assessment of state self-compassion is key, and the SSCS is a good tool for this end. This new scale is based on the original SCS, and uses 18 of its items that have been rewritten in the present moment language. Additionally, participants are also instructed to think about current painful or difficult situations. The SSCS has demonstrated evidence of validity and reliability, with state self-compassion being related to positive and negative affect, and adequately assessing experimental manipulations of self-compassion [62].

The purpose of this study is to test the psychometric properties of the Spanish version of the State Self-Compassion State–Long.

## 2. Materials and Methods

### 2.1. Design

The study has a longitudinal design. Four surveys of nursing students will be conducted during the years 2022–2025. Cross-sectional data of the first wave was used. This first assessment (time 1 or pre–internship assessment) took place during May 2022. The study has been reported using the Strengthening the Reporting of Observational Studies in Epidemiology (STROBE) Statement [68].

For the translation of the scale, we used the backward and forward translation process. First, the scale was translated into Spanish by two professional natives. Then, it was translated back into English by two native professionals. In the final version, no differences were found. The resulting Spanish version of the scale can be consulted in Table 1.

### 2.2. Participants

Students in the first year of the Nursing Degree from the University of Valencia and the University of the Balearic Islands (Spain) were encouraged to participate. In order to be included, the participants had to be nursing students, with no previous experience of internships. The students fulfilled the online questionnaire in approximately 20 min.

For the determination of sample size, the total population of nursing students of both universities was considered. For a population of students that was calculated to be N = 1880, with a confidence interval of 95% and an error limit of 5%, the number of elements of the sample to obtain was n = 320.

### 2.3. Instruments

The Spanish version of the State Self-Compassion State (SSCS–L) [62] includes 18 items, assessing six dimensions of self-compassion; self-kindness, common humanity and mindfulness would represent compassionate behaviors, whereas their opposite poles, self-judgement, isolation and over-identification, would represent the uncompassionate ones [4]. Each dimension is assessed with three items, which refer to a painful or difficult situation that the respondent is experiencing in that time. Items score in a five-point Likert-type scale, from 1 (not at all true for me) to 5 (very true for me).

Additionally, the Positive and Negative Affect Scale (PANAS) [69] was used to gather evidence of criterion-related validity. This is a 20-item self-reported measure of positive and negative mood, in which participants rate how they are feeling at the time using a series of adjectives (e.g., disgusted, irritable). Items score in a 5-point Likert-type scale, ranging from 1 (very slightly or not at all) to 5 (extremely). For the positive affect score, the mean of the 10 items representing positive adjectives is calculated, while for the negative one, the mean of the 10 items representing negative adjectives is calculated. This version of the PANAS has previously demonstrated adequate reliability, and was used in the original study of development and validation of the State Self-Compassion Scale–Long form (SSCS–L) [62]. In the original study [69], reliability estimates ranged from 0.860 to 0.900 for positive affect, and from 0.840 to 0.870 for negative affect; in this sample, estimates of reliability were 0.901 and 0.782, respectively.

### 2.4. Data Analysis

Firstly, descriptive statistics for the items of the scale, including means, standard deviations, and minimum and maximum scores, were calculated.

Secondly, construct validity was studied. For the study of the internal structure, competitive confirmatory factor analysis (CFA) was used. Three CFAs were specified and tested, following Lluch-Sanz et al.’s [42] work. Specifically, the tested models included:One-factor model. This model tested a general factor of self-compassion that explained the 18 items of the SSCS–L—that is, it tested a unidimensional structure.Two-correlated-factor model. This model hypothesized a two-correlated factor solution, a positive self-compassionate factor and a negative one.Six-correlated-factor model. This model hypothesized six correlated factors, including self-kindness, common humanity and mindfulness, which would represent compassionate behaviors, together with self-judgement, isolation and over-identification representing their opposite poles.

Details can be consulted in Figure 1.

To assess the model fit, several criteria were used: the chi-square statistic, the Comparative Fit Index (CFI), the Tucker-Lewis Index (TLI), the Standardized Root Mean Square Residual (SRMR), and the Root Mean Square Error of Approximation (RMSEA). Cut-off criteria to determine good fit were: CFI and TLI above 0.90 (better over 0.95) and SRMR or RMSEA below 0.08 (better under 0.06) [70]. For model comparison, as the competitive CFA were not nested, subjective criteria were used. That is, if a parsimonious model evidences adequate levels of practical fit, it is preferred over the more complex model. CFI differences (ΔCFI) lower than 0.010 [71] or 0.050 [72] were used as cut-off criteria. The model were estimated using the weighted least squares mean and variance adjusted (WLSMV) estimation method, according to the ordinal nature of the data and its non-normality [73,74,75].

Then, we gathered evidence of criterion-related validity. The SSCS–L was related to positive and negative affect. For this purpose, Pearson correlations were calculated. Additionally, sex differences in self-compassion were studied using *F* tests.

Finally, and in order to study the reliability of the scale, Cronbach’s alpha and the Composite Reliability (CR) Index were used. Values of Cronbach’s alpha and CR greater than 0.70 are considered adequate.

For the statistical analyses, SPSS version 26 (IBM, New York, NY, USA) [76] and Mplus version 8.4 (Muthén and Muthén, Los Angeles, CA, USA) [75] were used.

## 3. Results

### 3.1. Sociodemographic Characteristics of the Sample

The final sample was composed by 324 nursing students, of whom 84.0% were women. Their mean age was 22.61 (SD = 7.22) years old. 51.9% were University of Valencia students, and 48.1% were University of the Balearic Islands students. Most lived with their parents (43.2%) or with other students (30.2%). The majority of the students were not working at the time of the survey (70.7%). 14.5% were working as healthcare professionals. For more details, see Table 2.

### 3.2. Items’ Descriptive Statistics

Descriptive statistics were calculated for the items of the scale. Means ranged from 2.51 (item 18) to 3.85 (item 9). These can be consulted in Table 1.

### 3.3. Construct Validity

To study the internal structure of the scale, three were specified and tested. The one-factor and the two-correlated-factor model showed inadequate fit, whereas the six-correlated-factor model showed an excellent fit (see Table 3). Therefore, this latest model was retained as the best fitting model.

As displayed in Figure 2, the highest factor loading for the self-kindness dimension was item 13 (“*I’m being supportive toward myself*”), while for the self-judgement dimension it was item 10 (“*I’m being a bit cold-hearted towards myself*”), for common humanity it was item 9 (“*I’m remembering that there are lots of others in the world feeling like I am*”), for isolation it was item 18 (“*I’m feeling all alone right now*”), for mindfulness it was item 17 (“*I’m keeping things in perspective*”), and for over-identification it was item 2 (“*I’m obsessing and fixating on everything that’s wrong*”). Although all factor loadings resulted in statistical significance, a very small value was found for item 8 (“*I’m getting carried away with my feelings*”) in the dimension of over-identification.

### 3.4. Criterion-Related Validity

In the third place, we gathered evidence of criterion-related validity by relating the SSCS–L scores with levels of positive and negative affect. The Pearson correlations, displayed in Table 4, showed positive relations between the positive dimensions of self-compassion and positive affect, and negative relations between the positive poles of self-compassion and negative affect. That is, higher levels of self-kindness, common humanity and mindfulness were positively related to positive affect and negatively related to negative affect. On the contrary, higher levels on the negative dimensions of self-compassion, which included self-judgement, isolation and over-identification, were related to higher levels of negative affect and lower levels of positive affect.

Regarding sex differences in self-compassion, statistically significant differences were found in the dimensions of self-kindness and mindfulness, with lower levels for women. In self-judgment, common humanity, isolation and over-identification differences between women and men were not statistically significant (see Table 4).

### 3.5. Reliability

To study the reliability of the SSCS–L scale, Cronbach’s alpha and the Composite Reliability Index were used. Reliability estimates of all the dimensions resulted as adequate, except for over-identification. This dimension showed poor reliability, both when estimated using Cronbach’s alpha and CR (see Table 5).

## 4. Discussion

The current research analyzed the psychometric properties of the Spanish version of the SSCS–L scale [62]. Evidence of construct, convergent and discriminant, and criterion-related validity was gathered, as well as reliability estimates. A forward–backward translation was conducted to ensure that the Spanish version measured the same features as the English version.

Our results pointed that the Spanish version of the SSCS–L scale exhibited excellent evidence of construct validity. Indeed, the competitive CFA models pointed the six-correlated-factor structure as the best representation of the data. This is in line with previous studies carried out with both the long and short forms of the Self-Compassion Scale (SCS). This is demonstrated in a variety of studies that reveal that a six-factor structure best reflects health service worker outcomes, such as Raes et al. [48], who found evidence of a six-factor structure, with a single higher-order factor of self-compassion, or García-Campayo et al. [60], who found this six-factor structure to be the solution for both SCS and SCS/SF in Spanish Health Service workers. The six-factor structure of self-compassion has also been identified amongst the Saudi nursing students Alabdulaziz et al. [77]. We know from the scientific literature that all six self-compassion dimensions can be improved by programed interventions. Some of the successful interventions tested, for example, are those used by Rao and Kemper [46], in which online training in Gratitude, Positive Word and Love-Kindness/Compassion Meditation showed a significant improvement in all the dimensions of self-compassion. Similarly, the study by Sansó et al. [78] evidenced significant improvements in all the dimensions of self-compassion after a mindfulness training program in palliative care professionals, except for the Common Humanity dimension.

Regarding criterion-related validity evidence, the results obtained in the current study were in line with self-compassion previous literature: higher positive affect is related to higher levels of self-compassion (in its positive meaning), and higher negative affect is related to lower levels (lower scores in self-kindness, common humanity and mindfulness, and higher ones in self-judgement, isolation and over-identification) [2,62,79,80]. It is worth noting the positive effect that self-compassion has emotions, since self-compassion not only replaces negative emotions, but also generates positive emotions through the acceptance of negative states [81]. On other hand, self-compassion attenuates the negative affective responses [82]. Indeed, some authors have pointed to the role of affect in self-regulation as an alternative explanation for the beneficial effect of self-compassion on health behaviors [66,82].

Regarding sex differences in self-compassion, lower levels of self-kindness and mindfulness were found for women. This is in line with previous research, which has pointed out that males had slightly higher levels of self-compassion than females [83,84]. However, it is noteworthy to mention that the sample of this study was mostly made up of women, and therefore men’s presence in the study was very small in size. Therefore, generalization of these results should be made with caution.

Finally, results on reliability pointed adequate reliability estimates for the dimensions of the Spanish version of the SSCS–L scale, except for the dimension of over-identification. This result is in line with Neff’s study [62]. A possible explanation of the failure of the over-identification dimension could be that this dimension is made up of three items; two of these are clearly negative, but the third is less pronounced, since it literally says: *I’m getting carried away with my feelings*. Therefore, some respondents may interpret this item in a positive way, contrary to the remainder of the dimension. It is precisely the only item with low reliability, and the culprit of such low reliability of the entire dimension. Future studies should delve into the meaning of this item, especially in samples that are not familiar with meditative techniques, such as the one in this study. For example, studies on content validity could be done, in order to delimitate the representativeness and adequacy of the items’ content to the evaluation of state self-compassion.

### Study Limitations

This study presents some shortcomings, including the small size of the sample, which can affect to the generalizability of the results obtained. Additionally, test-retest reliability estimates could not be provided. Another limitation regards the models tested, which do not include the vastness of proposed structures for the study of self-compassion. Finally, and as stated by Neff et al. [62], it should be noted that the SSCS–L scale measures the construct of self-compassion as defined by Neff [1,2], and therefore it cannot be used to assess other definitions of self-compassion. Future longitudinal studies testing the psychometric properties of the Spanish version of the SSCS–L scale in bigger, more representative samples should be welcomed.

## 5. Conclusions

The results of the present study show that the Spanish version of the SSCS-L is psychometrically valid, and is a reliable measure for the state self-compassion in nursing students. Self-compassion, which is a protective factor against stress and burnout, can be enhanced through the development of positive affect, and the reduction of negative affect. As self-compassion is positively related with empathy, compassion for others, effectiveness of care and compassionate care, it has become a key competency for healthcare professionals. As the professionals most vulnerable to suffer the negative effects of helping others with their suffering, nurses—and especially nursing students—need to have a valid and reliable tool that allows for periodic assessments of self-compassion, as one way to improve professionals’ well-being and to ensure compassionate care.

The global shortage of nurses, exacerbated by the health crisis caused by the COVID-19 pandemic, makes it even more necessary to work on all those aspects that can prevent processes such as burnout and compassion fatigue. Both of them can lead to premature abandonment of the profession. Introducing skills such as self-compassion and valuing them in undergraduate training when professionals are still in training is possibly the most effective time to do so. The State Self-Compassion Scale–Long (SSCS–L) is an essential tool for this purpose, as it will allow for an adequate assessment of self-compassion in experimental contexts. The results of the present study will facilitate the incorporation of soft skills such as self-compassion in the curriculum of nursing students.

## Figures and Tables

**Figure 1 ijerph-19-10174-f001:**
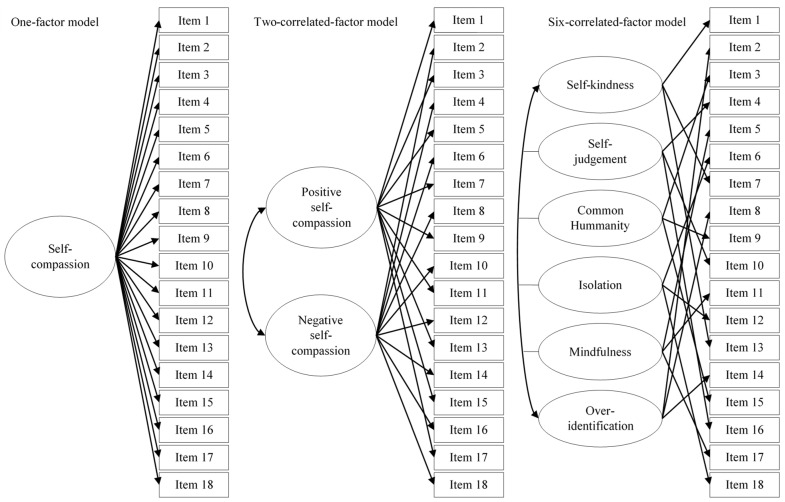
Competitive Confirmatory Factor Analyses for the SSCS–L.

**Figure 2 ijerph-19-10174-f002:**
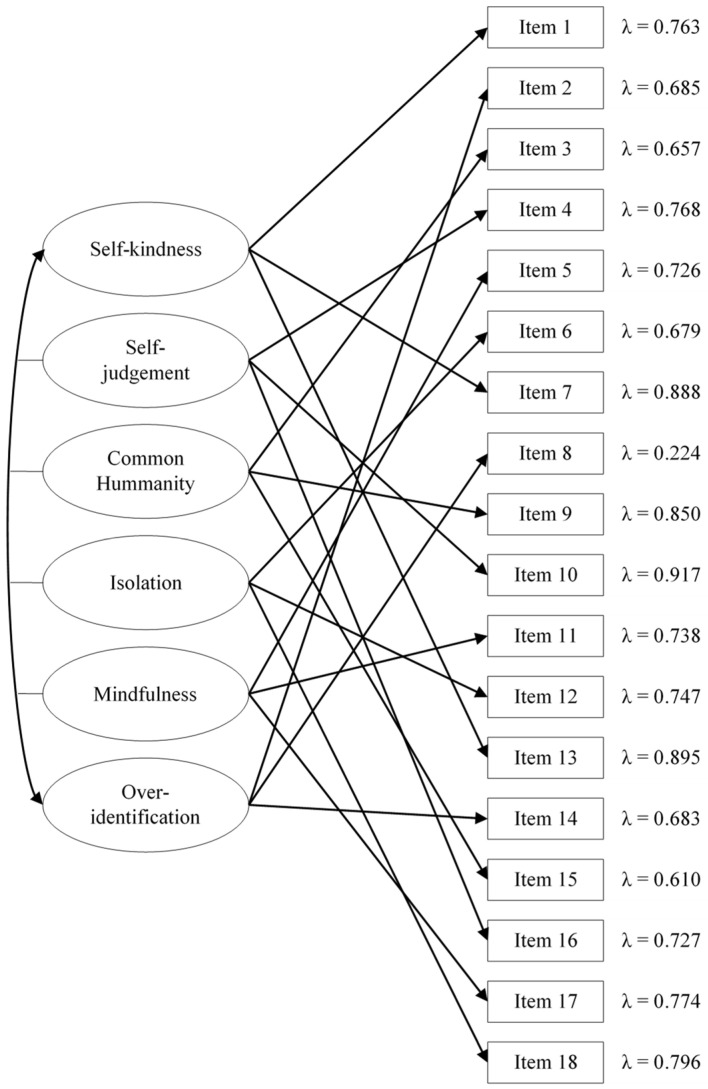
Analytical fit of the best fitting model for the SSCS–L. Notes: All factor loadings resulted statistically significant (*p* < 0.001). For the sake of clarity, standard errors are not shown.

**Table 1 ijerph-19-10174-t001:** Items and descriptive statistics for the SSCS–L.

Item *	M	SD	Min	Max
1. I’m giving myself the care and tenderness I need. *1. Me doy todo el cuidado y el cariño que necesito.*	2.966	1.033	1.00	5.00
2. I’m obsessing and fixating on everything that’s wrong. *2. Me obsesiono y me centro únicamente en las cosas negativas.*	3.011	1.070	1.00	5.00
3. I see my difficulties as part of life that everyone goes through. *3. Veo las dificultades que tengo como algo por lo que todo el mundo pasa en la vida.*	3.474	0.955	1.00	5.00
4. I’m being pretty tough on myself. *4. Estoy siendo muy duro/a conmigo mismo/a.*	3.413	1.065	1.00	5.00
5. I’m keeping my emotions in balanced perspective. *5. Veo mis emociones con perspectiva.*	3.202	0.958	1.00	5.00
6. I feel separate and cut off from the rest of the world. *6. Me siento apartado/a y desconectado/a del resto del mundo.*	2.886	1.215	1.00	5.00
7. I’m being kind to myself. *7. Estoy siendo amable conmigo mismo/a.*	3.072	1.082	1.00	5.00
8. I’m getting carried away with my feelings. *8. Me estoy dejando llevar por mis emociones.*	3.302	0.937	1.00	5.00
9. I’m remembering that there are lots of others in the world feeling like I am. *9. Tengo presente que hay muchas otras personas en el mundo que se sienten como yo.*	3.853	0.968	1.00	5.00
10. I’m being a bit cold-hearted towards myself. *10. Estoy siendo un poco cruel conmigo mismo/a.*	3.061	1.210	1.00	5.00
11. I’m taking a balanced view of this painful situation. *11. Abordo esta dolorosa situación de forma equilibrada.*	3.023	0.927	1.00	5.00
12. I feel like I’m struggling more than others right now. *12. Siento que me está costando más que a otros en este momento.*	3.159	1.122	1.00	5.00
13. I’m being supportive toward myself. *13. Estoy siendo comprensivo/a conmigo mismo/a.*	3.038	1.005	1.00	5.00
14. I’m blowing this painful incident out of proportion. *14. Estoy magnificando demasiado esta dolorosa situación.*	2.924	1.132	1.00	5.00
15. I’m remembering that difficult feelings are shared by most people. *15. Tengo presente que la mayoría de la gente experimenta las mismas sensaciones desagradables.*	3.650	1.019	1.00	5.00
16. I feel intolerant and impatient toward myself. *16. Siento que soy un/a intolerante y un/a impaciente conmigo mismo/a.*	3.080	1.138	1.00	5.00
17. I’m keeping things in perspective. *17. Veo las cosas con perspectiva.*	3.269	0.979	1.00	5.00
18. I’m feeling all alone right now. *18. Ahora mismo me siento muy solo/a.*	2.515	1.279	1.00	5.00

* Italics for the Spanish version of the SSSC–L.

**Table 2 ijerph-19-10174-t002:** Sample description.

Variable	Groups	M ^1^	SD ^2^
Age	-	22.61	7.22
**Variable**	**Groups**	**n**	**%**
Sex	Women	272	84.0
	Men	51	15.7
	Missing	1	0.3
University	University of Valencia	168	51.9
	University of the Balearic Islands	156	48.1
	Missing	0	0.0
Living	Alone	11	3.4
	With parents	140	43.2
	With other students	98	30.2
	With couple	36	11.1
	Others	39	12.0
	Missing	0	0.0
Job	Without job	229	70.7
	Part-time job	44	13.6
	Full-time job	51	15.7
	Missing	0	0.0
Healthcare professional	No	277	85.5
	Yes	47	14.5
	Missing	0	0.0

^1^ M = mean; ^2^ SD = standard deviation.

**Table 3 ijerph-19-10174-t003:** Goodness-of-fit indices for the estimated solutions for the SSCS-L.

Model	*χ^2^*	df	*p*	CFI	TLI	RMSEA (90% CI)	SRMR
One-factor CFA	1269.710	135	< 0.001	0.741	0.706	0.177 (0.169, 0.186)	0.119
Two-factor CFA	673.873	134	< 0.001	0.877	0.859	0.123 (0.114, 0.132)	0.076
Six-factor CFA	319.126	120	< 0.001	0.955	0.942	0.079 (0.068, 0.089)	0.050

**Table 4 ijerph-19-10174-t004:** Criterion-related validity results for the Spanish SSCS–L.

Variable	Positive Affect	Negative Affect	*F*	df_effect_	df_error_	*p*	M_women_	SD_women_	M_men_	SD_men_
Self-kindness	0.532 **	−0.404 **	10.522	1	264	0.001	2.949	0.930	3.436	0.793
Self-judgement	−0.333 **	0.345 **	0.000	1	261	0.994	3.180	0.979	3.178	0.910
Common humanity	0.253 **	−0.094 n.s.	0.015	1	262	0.901	3.651	0.769	3.667	0.804
Isolation	−0.448 **	0.357 **	0.245	1	262	0.621	2.830	0.953	2.909	1.064
Mindfulness	0.419 **	−0.324 **	6.311	1	262	0.013	3.112	0.782	3.432	0.710
Over-identification	−0.255 **	0.284 **	0.594	1	262	0.442	3.057	0.736	3.151	0.782

** *p* < 0.001; n.s. = not statistically significant.

**Table 5 ijerph-19-10174-t005:** Cronbach’s alpha, CR and bivariate correlations for the Spanish SSCS–L.

Variable	SK ^1^	SJ ^2^	CH ^3^	IS ^4^	MI ^5^	OI ^6^	Cronbach’s alpha	CR
Self-kindness	-						0.857	0.887
Self-judgement	−0.584 **	-					0.810	0.848
Common humanity	0.362 **	−0.032 n.s.	-				0.681	0.753
Isolation	−0.488 **	0.784 **	−0.270 **	-			0.729	0.786
Mindfulness	0.892 **	−0.442 **	0.610 **	−0.412 **	-		0.746	0.790
Over-identification	−0.504 **	0.998 **	−0.065 n.s.	0.939 **	−0.445 **	-	0.506	0.557

^1^ SK = self-kindness; ^2^ SJ = self-judgment; ^3^ CH = common humanity; ^4^ IS = isolation; ^5^ MI = mindfulness; ^6^ OI = overidentification; CR = composite reliability; ** *p* < 0.001; n.s. = not statistically significant.

## Data Availability

The data that support the findings of this study are available from the corresponding author upon reasonable request.

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
