# Peer review of "The Spanish Version of the State Self-Compassion Scale–Long Form (SSCS–L): A Study of Its Validity and Reliability in a Sample of Nursing Students"

_ijerph, 2022, doi:10.3390/ijerph191610174_

Round 1

Reviewer 1 Report

Dear authors,

Two small remarks:

-              * Gender issues were nor even approached – are women more present in nursery programmes? how is the sample regarding gender representativity? are men’s results any different?... This might be important as self-care is a social feature that tend to be more associated to traditional female social roles.

-             * What the purposes of longitudinal study would be?

Author Response

REVIEWER:

Dear authors,

Two small remarks:

- Gender issues were nor even approached – are women more present in nursery programmes? how is the sample regarding gender representativity? are men’s results any different?... This might be important as self-care is a social feature that tend to be more associated to traditional female social roles.

ANSWER:

Gender issues have been addressed. We have studied sex differences in self-compassion using F tests. This information has been included in the Analysis section, in the Results, and in the Discussion section. Statistically significant differences were found in the dimensions of self-kindness and mindfulness, with lower levels for women. In self-judgment, common humanity, isolation, and over-identification differences between women and men were not statistically significant. These results have been included in Table 4. Regarding the discussion of these results, we have included the following information:

“Regarding sex differences in self-compassion, lower levels of self-kindness and mindfulness were found for women. This is in line with previous research, which has pointed out that males had slightly higher levels of self-compassion than females [83,84]. However, it is noteworthy to mention the sample of this study was mostly formed by women, and therefore men group in the study was very small in size. Therefore, generalization of these results should be made with caution.”

Regarding gender representativity of the sample, the sample obtained in the study is a clear representation of gender distribution of nurses students. For example, data from the University of Valencia pointed out that students enrolled in the first course during the 2021-2022 school year to the Degree in Nurse (Annual UV Data Book, 2022; https://www.uv.es/uvweb/analysis-programming-service/en/statistics-indicators/annual-uv-data-book/annual-uv-data-book-1285868428356.html) were 227 women (84.07%) and 43 men (15.93%). This distribution is almost identical to our study sample, in which 84.0% of the participants were women.

REVIEWER:

-  What the purposes of longitudinal study would be?

ANSWER:

As this study is a part of a bigger project, the reason to the longitudinal nature obey to the study design of the project, instead of to a specific purpose of the current study. That is, data of this study are cross-sectional. However, with the longitudinal project we will be able to give answers to some of our shortcomings, such as the absence of test-retest reliability estimates.

Reviewer 2 Report

Dear Author,

It is delightful to have the opportunity to review “The Spanish version of the State Self-Compassion Scale – Long form (SSCS–L): A study of its validity and reliability in a sample of nursing students”

I hope my opinions will contribute to the publication of the high-quality paper.

1)      Original scale: Did the author authorize its use?

2)      Table 2 – why is there a line separating the groups of the variable Living?

3)      Table 2 is not placed in the text near the first time it was cited.

4)      Line 183 - Review the title in table 3 - This is a table. Tables should be placed in the main text near to the first time they are cited.

5)      Table 3 – What is Title 1? Clarify.

6)       Lines 129-130 – What was the alpha Cronbach of the original scale?

7)       Lines 259-261 - Future studies should delve into the meaning of this item and the dimension in general. What suggestions do you propose to deepen the sense of this item?

8)      I believe it is important to clarify why the sample just includes students from the first year of the nursing degree. Why was it so important that the participants didn´t have previous experiences in internships?

I suggest:

1)      Line 18 - changes wellbeing for well-being.

2)      Table 2 – SD equal to 0.0 change to NA, not applied because there were no values to measure SD.

3)      Not job (no trabajo) maybe change to Without job (sin trabajo)

4)      Line 283 – change COVID to COVID-19

Finally, congratulations on the work.

Best regards,

Author Response

REVIEWER:

Dear Author,

It is delightful to have the opportunity to review “The Spanish version of the State Self-Compassion Scale – Long form (SSCS–L): A study of its validity and reliability in a sample of nursing students”

I hope my opinions will contribute to the publication of the high-quality paper.

1)      Original scale: Did the author authorize its use?

ANSWER:

The State Self-Compassion Scale, as the original Self-Compassion Scale, is of free use and no authorization for its use was needed. However, we wrote the author informing of our study. 

REVIEWER:

2)      Table 2 – why is there a line separating the groups of the variable Living?

ANSWER:

This was a mistake. We have changed it.

REVIEWER:

3)      Table 2 is not placed in the text near the first time it was cited.

ANSWER:

We have changed Table 2 placement according to the reviewer’s suggestion.

REVIEWER:

4)      Line 183 - Review the title in table 3 - This is a table. Tables should be placed in the main text near to the first time they are cited.

ANSWER:

Title of Table 3 has been reviewed. It now says “Goodness-of-fit indices for the estimated solutions for the SSCS-L”.

We have also changed Table 3 placement according to the reviewer’s suggestion.

REVIEWER:

5)      Table 3 – What is Title 1? Clarify.

ANSWER:

“Title 1” was a mistake, we forgot to change the title. We have now done it and change it for “Model”, which is the content of the first column of Table 3.

REVIEWER:

6)       Lines 129-130 – What was the alpha Cronbach of the original scale?

ANSWER:

According to the reviewer suggestion, we have included the estimates of reliability in the original study:

“In the original study [69], reliability estimates ranged from .860 to .900 for positive af-fect, and from .840 to .870 for negative affect; in this sample, estimates of reliability were .901 and .782, respectively.”

REVIEWER:

7)       Lines 259-261 - Future studies should delve into the meaning of this item and the dimension in general. What suggestions do you propose to deepen the sense of this item?

ANSWER:

We advocate for the study of content validity, as we believe this could be an appropriate tool to figure out if this was a specific sample problem, or instead it is a problem of item’s content representativeness. We have included this in the Discussion section:

“For example, studies on content validity could be done, in order to delimitate the representativeness and adequacy of the items’ content to the evaluation of state self-compassion.”

REVIEWER:

8)      I believe it is important to clarify why the sample just includes students from the first year of the nursing degree. Why was it so important that the participants didn´t have previous experiences in internships?

ANSWER:

As this study is a part of a bigger project, the reason to only include first year students obey to the study design of the project, instead of to a specific purpose of the current study. That is, there is not an explanation that participants did not have previous experiences in internships other that the work is part of a longitudinal project that will follow students during the four years of the Degree in Nursing, and, as this is the first year of the project, we have only gathered data from the first wave (that is, the first year of study). However, as this is a fact (participants did not have previous experiences in internships) and an exclusion criteria, we have include this information in the text.

REVIEWER:

I suggest:

1)      Line 18 - changes wellbeing for well-being.

ANSWER:

Done.

REVIEWER:

2)      Table 2 – SD equal to 0.0 change to NA, not applied because there were no values to measure SD.

ANSWER:

We believe the reviewer refers to the last column of Table 2. This has two subheadings; the first one, referring to SD, and the second one (row 3) referring to percentage (%). We have checked the Table and there is no SD value equal to 0.0, but this value is for %. Therefore, 0.0 values have been retained. Please let us know if we have not correctly checked the values.

REVIEWER:

3)      Not job (no trabajo) maybe change to Without job (sin trabajo)

ANSWER:

Done.

REVIEWER:

4)      Line 283 – change COVID to COVID-19

ANSWER:

Done.

REVIEWER:

Finally, congratulations on the work.

Best regards,

ANSWER:

Thank you very much.